# COMP-LTL: TEMPORAL LOGIC PLANNING VIA ZERO-SHOT POLICY COMPOSITION

## ABSTRACT

This work develops a zero-shot mechanism, Comp-LTL, for an agent to satisfy a Linear Temporal Logic (LTL) specification given existing task primitives trained via reinforcement learning (RL). Autonomous robots often need to safely and deterministically satisfy spatial and temporal goals that are unknown until run time. Prior work on learning policies to execute an LTL task incorporates the specification into the learning process, requiring retraining or fine-tuning if the specification changes. We present a more flexible approach–to create a pipeline to deterministically choose an execution set of composable safe task primitive policies that can be used to satisfy arbitrary LTL specifications without retraining or fine-tuning. Safe task primitives can be learned offline using RL with a reward function focused on penalizing unsafe actions and combined using Boolean composition at deployment. We focus on creating and pruning a transition system (TS) representation of the environment in order to solve for deterministic, non-ambiguous, and feasible solutions to LTL specifications given an environment with multiply-labeled regions and a set of safe task primitive policies. Our pruned TS is deterministic, contains no unrealizable transitions, and is sound. Combining the TS with the safe pretrained task primitives produces a sequence of composed policies that are guaranteed to deterministically satisfy an LTL specification. Training on a base set of safe tasks and composing at run time reduces total training time compared to non-composition approaches and has negligible processing time at run time. We verify our approach via simulation in grid-based and continuous environments, and compare it to other state of the art approaches, showing that Comp-LTL is safer, more adaptable, and quicker at satisfying unseen specifications at runtime.

## 1 INTRODUCTION

A major goal in autonomous systems is the deployment of robots that are capable of executing tasks that are time-varying, interdependent, and otherwise complex. One approach to addressing such complex task executions is planning with linear temporal logic (LTL) (Baier & Katoen, 2008; Kress-Gazit et al., 2018). LTL allows a user to specify tasks with complex temporal and inter-task relationships. A major strength of this approach is the focus on correct-by-construction algorithms that are capable of planning for an arbitrary formula specified by a user. However, many associated planning approaches require reliable task models in order to guarantee satisfaction of an LTL specification (Kress-Gazit et al., 2018; Belta et al., 2017).

Some works, such as Reward Machines (RM) (Icarte et al., 2018), use an automaton to learn policies, but they incorporate the specification into the learning process; therefore, they require retraining when provided with a new specification (Cai et al., 2023; Li et al., 2019). LTL-Transfer (Liu et al., 2024) is a zero-shot LTL solution that trains on transitions on a Büchi automata for a specification, and for new specifications, transfers the transitions to the respective Büchi automata. Although LTL-Transfer adheres to explicit safety in the specification, the zero-shot solution is constrained to transitions already explored during the training pass, whereas we desire a more broadly applicable solution.

A closely related approach is Skill Machines (SM) (Nangue Tasse et al., 2024), which leverages prior work on zero-shot composition (Nangue Tasse et al., 2020) to satisfy a proposition on a reward machine. While changing the specification does not require re-training from scratch, it nonetheless requires fine-tuning of the policies to guarantee satisfaction.

Other works reduce training time for multi-task RL, such as LTL2Action (Vaezipoor et al., 2021), or generalization for generating latent representations of a goal, such as (León et al., 2022), but do not provide a guaranteed zero-shot solution. Similarly, works that focus on zero-shot specification adherence, such as (Qiu et al., 2023) and (Jackermeier & Abate, 2025), must train on every goal and do not consider regions with multiple labels since their policies do not support composition, and, like most other work in the area, do not focus on safety guarantees. Our approach, on the other hand, focuses on providing guarantees possible for zero-shot transfer of safe policies and the required training and planning steps to claim those guarantees. Table 1 highlights the difference of our approach from prior approaches. A comprehensive review of related work is included in Appendix B.

| Method | Satisfies Given Specification | Skill Primitive Composition | Implicit Safety Guarantee | Generalization to Unseen Specifications |
|---|---|---|---|---|
| Comp-LTL | ✓ | ✓ | ✓ | Zero-shot |
| SM | ✓ | ✓ | | Zero shot possible, few-shot guaranteed |
| RM | ✓ | | | |
| LTL2Action | ✓ | | | Zero-shot for limited length specifications |
| LTLTransfer | ✓ | | | Zero-shot for previously observed transitions |

Table 1: Capabilities of Our Method, Comp-LTL, and Related Works
(Nangue Tasse et al., 2024; Icarte et al., 2018; Vaezipoor et al., 2021; Liu et al., 2024)

To better underscore our contributions, consider an example with three regions shown in Figure 1: 1) $W, R$; 2) $W, C$; 3) $R$, representing a waste ($W$) dump site in a residential ($R$) area, a waste ($W$) dump site in a commercial ($C$) area, and residential area ($R$), respectively, with the $R$ region in front of (but not completely surrounding) the $W, C$ region. If we provide the LTL specification $F(W)$, Comp-LTL will go to either the residential waste site or the commercial waste site. Our path to the $W, R$ site is dashed green, and the path to the $W, C$ path is dotted blue.

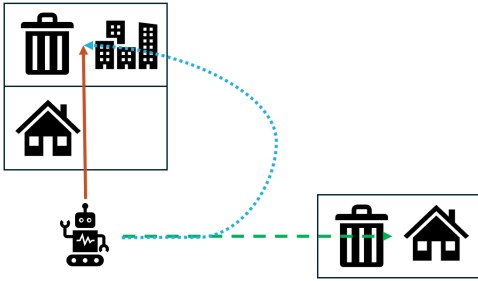

Figure 1: Paths the agent could take for the specification $F(W)$. Comp-LTL: the dotted blue (safe) or dashed green. Other methods: the solid orange (unsafe) or dashed green.

If a truck were carrying hazardous material, it cannot safely pass through the residential ($R$) neighborhood on the path to the commercial waste site ($W, C$). Our approach is the only approach to avoid the region implicitly, so it will successfully avoid $R$ with the specification $F(W \wedge C)$, as seen via the blue dotted line, whereas the other approaches need to explicitly train on an $R$ region and include it in their specification to avoid it, as $F(W \wedge C) \wedge G \neg R$. Without this explicit avoidance criteria, they would follow the dangerous orange solid path. We compose policies associated with each individual label, reducing the total number of policies needed to be trained prior to run time. Most other approaches would need to train on every possible combination of labels (e.g., $W$, $C$, and $W \wedge C$) to mirror our behavior.

In this work, we propose Comp-LTL, a framework for finding a satisfying solution for an environment and specification regardless of the exact environment, specification, or policies. Inspired by Kloetzer & Belta (2008) and recent work in zero-shot Boolean Composition (Nangue Tasse et al., 2020) (BC), we observe that compositional approaches allow us to satisfy Boolean constraints on automaton representations of LTL specifications. We leverage the prior work on safety-aware Boolean compositions of primitive policies to ensure the solution can be run zero-shot (Leahy et al., 2024), and that the satisfying word can be achieved in the environment. Figure 2 shows our approach, Comp-LTL.

The specific contributions of this work are the following:

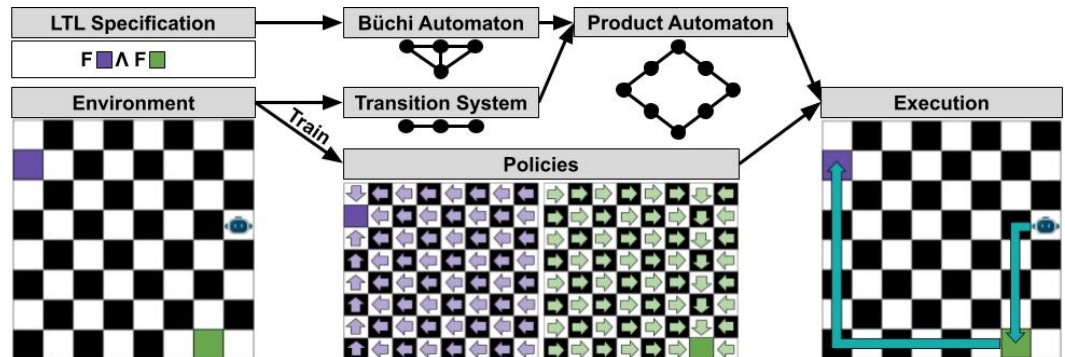

Figure 2: Comp-LTL training and path execution pipeline.

1. We develop a method for abstracting a geometric representation of an environment into a transition system (TS) with transition labels representing feasible Boolean combinations of tasks to transition between multiply labeled regions;

2. We resolve nondeterminism in the transitions enabled by the Boolean composition of primitive task policies; and

3. We demonstrate that this representation allows zero-shot satisfaction of LTL specifications at run time, and the resulting behavior is inherently safe without adding specific safety criteria into the specification.

Our goal is to produce a behavior sequence that is guaranteed to satisfy an LTL specification. Neither embedding safety in the policies nor pruning the TS on its own will accomplish this. We support our theoretical results with case studies in simulation and comparison to other approaches.

## 2 BACKGROUND AND PROBLEM FORMULATION

We consider an agent moving in a planar environment according to a high-level mission description. The agent's environment is $E \subseteq \mathbb{R}^n$. The environment contains non-intersecting regions $R \subseteq E$. We define a region labeling function $L : \mathcal{R} \to 2^{\Sigma}$, where $\mathcal{R}$ is the set of all labeled regions and $\Sigma$ is the set of atomic propositions (AP). Each region can be labeled with multiple APs (*multiply labeled*).

**Assumption 1** *We assume the environment is deterministic, so we further model the agent's environment as a deterministic labeled Markov decision process (MDP).*

Since our pruned TS is deterministic, following our method in Section 3.2, it is consistent with our assumption of a deterministic labeled MDP, defined by the tuple $(\mathcal{S}, \mathcal{A}, \rho, r)$, where $\mathcal{S}$ is the state space. The labeling function of an MDP includes a set of goals: $\mathcal{G} \subseteq \mathcal{S}$. An *execution* of a labeled MDP is the word $\tau \in (2^{\sigma})^{\omega}$, consisting of the sequence of labels corresponding to the regions the agent visits. To transform an agent's interaction with the environment into a set of labels, we *project* an execution onto the set of associated AP labels: $\upharpoonright_L : \mathcal{S} \times 2^{\Sigma} \to 2^{\Sigma}$.

The agent's task is specified using linear temporal logic (LTL) (Baier & Katoen, 2008). LTL includes Boolean operators, such as AND ($\wedge$), OR ($\vee$), and NOT ($\neg$), along with time based operators *eventually* ($\Diamond$), *always* ($\Box$), and *until* ($\mathcal{U}$). The formal syntax of LTL in Backus–Naur form is

$$\phi ::= \top | \sigma | \neg \sigma | \phi_1 \vee \phi_2 | \phi_1 \wedge \phi_2 | \phi_1 \mathcal{U} \phi_2 | \Diamond \phi_1 | \Box \phi_1 , \tag{1}$$

where $\sigma \in \Sigma$ is an atomic proposition, and $\phi$, $\phi_1$, and $\phi_2$ are LTL specifications (Baier & Katoen, 2008).

Due to space constraints, we do not describe the semantics of LTL here, but provide a brief intuition. A execution sequence $\tau \in (2^{\Sigma})^{\omega}$ satisfies a specification $\phi$ (written $\tau \models \phi$) if the sequence matches the properties specified by $\phi$. For example, if $\phi = \Diamond \sigma$ ("eventually $\sigma$"), $\tau \models \phi$ if $\sigma$ occurs at some point in $\tau$. Similarly, $\Box \sigma$ ("always $\sigma$") requires that $\sigma$ appear at every point in $\tau$. Interested readers are directed to Baier & Katoen (2008) for more details on the semantics of LTL. Importantly, off-the-shelf

software, such as SPOT (Duret-Lutz et al., 2022), can automatically translate LTL specifications into Büchi automata. Furthermore, each transition on such automata can be described by a Boolean combination of atomic propositions.

We assume the environment transition model is unknown to the agent, so we use reinforcement learning (RL) to learn policies for the agent to execute tasks. RL is a branch of machine learning that maps states to actions in order to maximize a numerical reward signal (Sutton & Barto, 2018).

To facilitate satisfaction of temporal logic objectives, we leverage prior work (Leahy et al., 2024) on safety-aware task composition to train policies for a given set of tasks. Other compositional works consider reachability-only (RO) semantics (Nangue Tasse et al., 2020). Negating a task in RO context means an agent will not terminate in the region associated with the negated task, but it could pass through that region. Negating a task in safety-aware context (i.e., within the primitive policy) means that the agent will always avoid the region associated with the negated task, which aligns with LTL requirements. If safety is not included at the task level, negation/avoidance cannot be ensured at run time, only that something will be reached.

To train safety-aware policies Leahy et al. (2024) proposed a method for learning policies that have "minimum-violation" (MV) safety semantics. For $\tau$, let the number of positions in the word with non-empty symbols be denoted $|\tau|$ and the set of symbols in the last position of the word be denoted $\tau_f$. Then, for a Boolean formula $\varphi$, we define MV semantics.

**Definition 1** *Minimum-violation (MV) Path: A word $\tau$ is a minimum-violation path if $|\tau| > 1$ and $\tau_f \models \varphi$ and there is no word $\tau'$ such that $|\tau'| < |\tau|$ (Leahy et al., 2024).*

Intuitively, an MV path: 1) terminates in a state that satisfies a Boolean formula; 2) if possible, visits no additional labeled states; and 3) if not possible, visits the fewest additional labeled states. Additional details on safety-aware MV semantics can be found in Appendix D.2.

In order for a policy to enforce this behavior, any label generated that does not satisfy the current task is given a penalty. To enforce the multiple levels of behaviors to be avoided, the rewards are structured hierarchically, with less bad rewards for passing through an unsafe state than terminating in an unsafe state (Leahy et al., 2024). Additional training details, including Table 5 with the reward hierarchy, is included in Appendix H.3.

We employ Boolean task algebra in order to perform task conjunction $\wedge$ over these MV policies (Nangue Tasse et al., 2020). The conjunction of two tasks is performed by taking the minimum of their individual Q-value functions:$Q_{1 \wedge 2}(s, g, a) = min[Q_1(s, g, a), Q_2(s, g, a)]$. We refer the reader to Appendix D.1 for more information. Boolean task algebra allows the agent to use a pretrained set of primitive tasks to expand the number of tasks that can be achieved with no additional learning by expressing the additional tasks as a Boolean expression over the original task primitives.

**Problem 1** *Given a set of labels $\Sigma$, an environment $E$ labeled from $\Sigma$, and safety-aware primitive policies trained to achieve tasks $\sigma \in \Sigma$ according to Leahy et al. (2024), select and compose primitive policies such that the policies can be used to satisfy an LTL specification $\phi$ over $\Sigma$ without additional training.*

## 3 Technical Approach

To solve Problem 1 we introduce a novel policy-aware environment abstraction as described below. First, we create a transition system (TS) (Belta et al., 2017) that captures both the topology of the environment as well as the policies for each primitive task to move the agent between regions. Constructing such a TS is conservative and can introduce ambiguity and non-determinism. To that end, we identify 3 cases for pruning the edge labels to remove non-determinism due to the reliance on task composition. The resulting TS can be used for planning to satisfy an LTL specification in the standard method (Belta et al., 2017), while accurately capturing the behavior created by the RL policies.

## 3.1 GENERATING THE TRANSITION SYSTEM

To facilitate reasoning about satisfaction of an LTL specification, we abstract the environment as a TS. A TS describes the discrete behavior of a system via states and transitions and is formally defined as follows.

**Definition 2** *A transition system (TS) is a tuple, $TS = (S, Act, \rightarrow, I, \Sigma, L)$, where $S$ is a finite set of states; $Act$ is a finite set of actions; $\rightarrow \subseteq S \times Act \times S$ is a transition relation; $I \in S$ is an initial state; $\Sigma$ is a set of atomic propositions; and $L : S \rightarrow 2^\Sigma$ is a labeling function.*

To create the initial TS, each region is instantiated as a state, and adjacent regions are connected by transitions; this captures the topology of the environment. Algorithm 1 in Appendix E.1 generates the transition labels. In planning- and control-based approaches, it is typical to assume that an agent can travel between any adjacent regions. For example, Fig. 3a shows an environment and a corresponding TS (3b). In a planning framework, an agent may choose which of the regions labeled $a$ to visit. Using our RL approach, however, for an agent in the unlabeled region $q_2$, executing a policy corresponding to $a$ may cause the agent to visit $q_1$, $q_3$, $q_5$, or $q_6$, since the transition function is unknown. We introduce a pruning process to model and resolve such ambiguities.

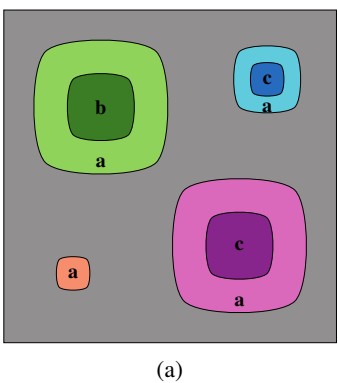
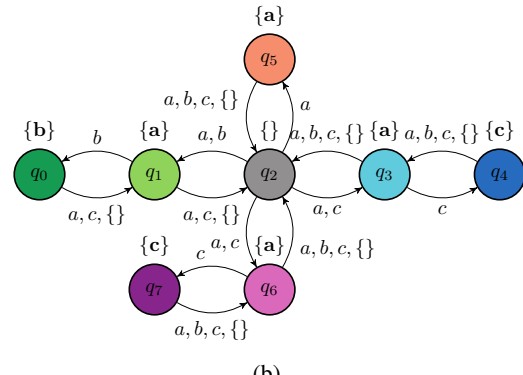

(a)                                        (b)

Figure 3: (3a) Example of an environment with distinct regions labeled with $\sigma \in \Sigma$. (3b) A corresponding unpruned TS created from the environment in Fig. 3a. Region colors from the environment in Fig. 3a correspond to the colors of the state nodes. State labels appear in **bold** above each state. Transition labels appear in *italics* adjacent to the transition arrows and correspond to task policies that enable a transition.

To resolve this non-determinism, we propose a method for pruning the TS. To prune, we will remove transitions and policies in transition labels that introduce non-determinism, by checking for the specific cases of: 1) Equivalency; 2) Ambiguity; and 3) Feasibility; with the methods for mitigating these cases later described in Sec. 3.2.

## 3.2 TRANSITION SYSTEM PRUNING

When we follow the procedure outlined in Sec. 3.1, we capture how states are connected, but the resulting TS state and transition labels can introduce non-determinism. To mitigate such problems, we introduce a TS pruning method, which removes symbols from transition labels.

We propose the following theorems about our method, please refer to Appendix G for the complete proof sketches.

**Theorem 1** *The resulting pruned TS from Sec. 3.2 is deterministic.*

**Theorem 2** *The resulting pruned TS from Sec. 3.2 contains no unrealizable transitions.*

**Case 1: Equivalent States** *If a bisimulation $TS_\approx$ exists for $TS$, reduce the total number of states by using $TS_\approx$.* The TS contains multiple branches from a parent state which contain the same state

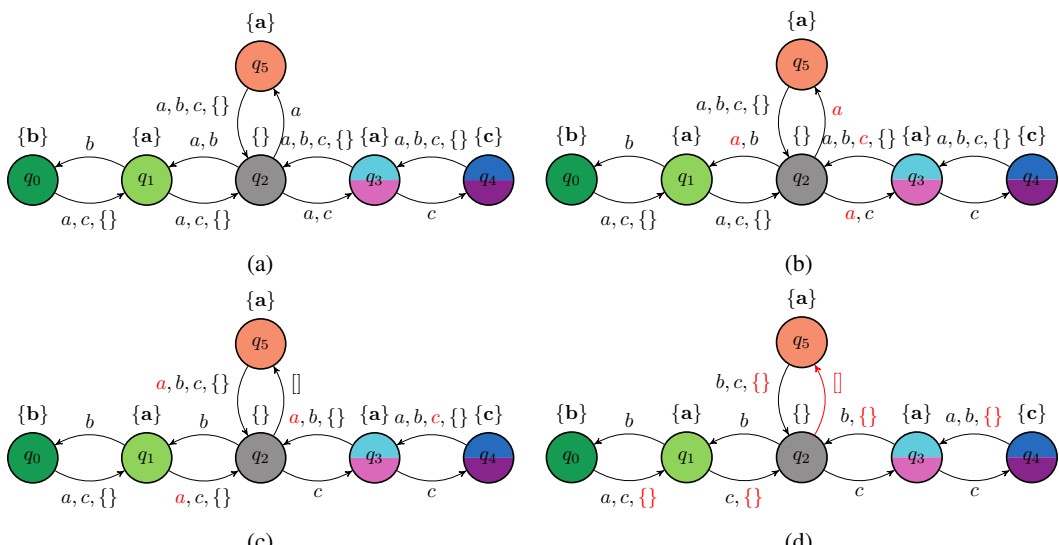

Figure 4: The TS pruning process. State labels appear above each state. Transition labels correspond to task policies that enable a transition. 4a The $q_3$, $q_4$ and $q_6$, $q_7$ branches are combined via `case1`. 4b Red label symbols are removed via `case2`. 4c Red label symbols are removed via `case3`. 4d Red label symbols are removed via `emptyCleanup`

and transition labels. In order to simplify the TS, we employ Bisimulation, which uses observational equivalence to reduce the TS (Belta et al., 2017). In a TS where multiple branches from a parent state contain the same state and transition labels, taking a policy from one of the symbols shared on the transition label could take the agent to any of the duplicate child state regions. The branches are then observationally equivalent, so we can reduce the TS to a bisimulation TS. Please refer to Appendix E.2.1 for a more detailed explanation.

Figure 4a highlights the changes in the TS after `case1` is executed. Algorithm `case1` identifies that the TS has equivalence classes. $q_2$ has branches that are equivalent. The two equivalent branches are 1) the branch containing $q_3$ and $q_4$ and 2) the branch containing $q_6$ and $q_7$. In Fig. 4a, the two equivalent branches get combined into the branch containing $q_3$ and $q_4$, and this resulting TS is $TS_\approx$.

**Case 2: Ambiguous Transitions** *If any outgoing transitions from a state share a symbol in the transition label, only keep the symbol in the transition with the least distance to the state labeled with the shared symbol, according to MV semantics. If all the distances to the state that is labeled with the shared symbol are the same, remove the symbol from all the transition labels of the state.* If a state has multiple transition labels that contain the same symbol, it is uncertain which transition will be followed when the corresponding policy is executed. We seek a method that is zero-shot, so we perform no additional checks or training on the policy to see how it would behave if run in the state region; therefore, we wish to keep at most once outgoing transition labeled with that symbol. Algorithm 3 in Appendix E.2.2 shows the procedure for `case2`.

Figure 4b highlights the changes in the TS after `case2` is executed. No labels $a$ are kept on outgoing transitions from $q_2$, because MV semantics cannot distinguish them. The label $c$ is removed from the transition linking $q_3$ to $q_2$, because MV semantics will result in an agent transitioning from $q_3$ to $q_4$ under a policy associated with task $c$.

**Case 3: Ineffectual Transitions and Feasibility** *If a state shares the same label as any outgoing transitions, remove the label from those transitions.* This case only arises when there are multiple states containing the same symbol label during the initial TS creation. Each duplicate state will have an outgoing transition label containing the same symbol as its own label, to get to the other states that share the same symbol label. We prune the symbol from the outgoing transition labels as running the policy for generating a symbol while already in the region that produces the symbol will not cause

the agent to transition out of its current state. Therefore, since the state does not change, the symbol on the label is ineffectual.

Figure 4c highlights the changes in $TS_\approx$ after `case3` is executed. State $q_1$'s label is $a$, and the transition from $q_1$ to $q_2$ contains $a$, so $a$ is removed from that transition. The same logic applies to the other highlighted labels.

### 3.3 Product between Transition System and Büchi Automaton

**Theorem 3** *Satisfying an LTL specification using product construction with our pruned $TS_\approx$ is sound, meaning there are no false positives.*

Given a fully pruned $TS_\approx$ with labels from $\Sigma$, we create a Büchi automaton using a an LTL specification $\phi$ over $\Sigma$. Importantly, transitions in the Büchi automaton are enabled by Boolean combinations of elements of $\Sigma$. Labels in $E$ and $TS_\approx$ model how an agent satisfies those Boolean combinations. Hence, we can then construct a Cartesian product between $TS_\approx$ and the automaton, preserving the transition labels from $TS_\approx$. The resulting product automaton (PA) can be used with typical off-the-shelf solution methods to find a satisfying sequence (Belta et al., 2017). Appendix F includes a complete description of PA generation and Appendix G includes the proof sketch.

## 4 Results

### 4.1 Simulation Case Study

To demonstrate our logic, we used three different environments: an office world environment based on Icarte et al. (2018), a high-dimensional video game environment based on Nangue Tasse et al. (2020), and a continuous 3D physics simulation Gronauer (2022). Our TS and PA are constructed using NetworkX (Hagberg et al., 2008) and a modified version of LOMAP [1]. Our Büchi automaton is created using SPOT (Duret-Lutz et al., 2022). The composition of policies is performed zero-shot via the method of Leahy et al. (2024). Appendix H.3 includes additional environment and training information.

**Office World Environment** The office world environment is a grid-world with symbols from the set of propositions {A (lobby:☎), B (labeled office), C (labeled office), D (breakroom:☺), E (mailroom:✉), F (coffee:☕), G (printer room:🖨), n (plant:🌿)}. Each grid cell may contain any symbol from the set of propositions. To expand on the typical office world environment, we trained symbols on each of the 4 quadrants of the map, which allows the user to be more specific about which symbols can satisfy the specification. These create the extra propositions $\{\uparrow, \downarrow, \leftarrow, \rightarrow\}$. We trained all 12 policies using tabular Q-learning with an MV reward structure (Leahy et al., 2024).

Given the LTL specification $\Diamond B \wedge \Diamond \text{🌿}$, our accepted word $\tau$ is $[B, \text{🌿}]$. Those respective policies are shown in Figure 5b and Figure 5a, and the path for those policies executed is shown in Figure 5c.

Next, we demonstrate compositionality in the office world environment. For example, there are two ☕symbols in the environment. If the user wants to specify the ☕in the top half of the office, they can specify $\Diamond \text{☕} \wedge \uparrow$. Figure 6a shows the composed values and policies for $\text{☕} \wedge \uparrow$. Comp-LTL computes that for specification $\Diamond C \wedge \Diamond(\text{☕} \wedge \uparrow) \wedge \Diamond(\text{🌿} \wedge \downarrow)$ the accepted word $\tau$ is $\tau = [\text{☕} \wedge \uparrow \wedge \leftarrow, \text{🌿} \wedge \downarrow \wedge \leftarrow, C \wedge \uparrow \wedge \rightarrow]$, with the path from the green start shown in Figure 6b.

**Video Game Environment** The video game environment is a grid-world. Each cell may contain an object characterized by a color and a shape from the set propositions {w (white), b (blue), p (purple), ● (circle), ■ (square)}. These traits can be composed in a Boolean fashion, e.g., $\blacksquare := b \wedge \blacksquare$. A policy is trained for each proposition using the Deep Q-Learning (DQL) RL methodology with MV safety semantics (Leahy et al., 2024).

The first example is shown in Fig. 7. For the LTL specification $\Diamond \blacksquare$, "eventually square", Comp-LTL produces the shortest word $\tau$ [$\blacksquare$], which translates to the Boolean composition policy $\pi_\blacksquare := \pi_b \wedge \pi_\blacksquare$.

---

[1]https://github.com/wasserfeder/lomap

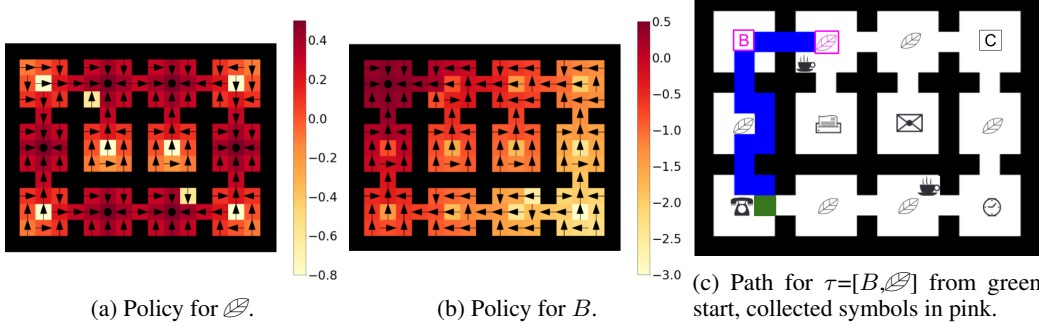

(a) Policy for 🍃.

(b) Policy for $B$.

(c) Path for $\tau=[B,$🍃$]$ from green start, collected symbols in pink.

Figure 5: Policies and path of the accepted word $\tau$ for LTL specification $\Diamond B \wedge \Diamond$🍃.

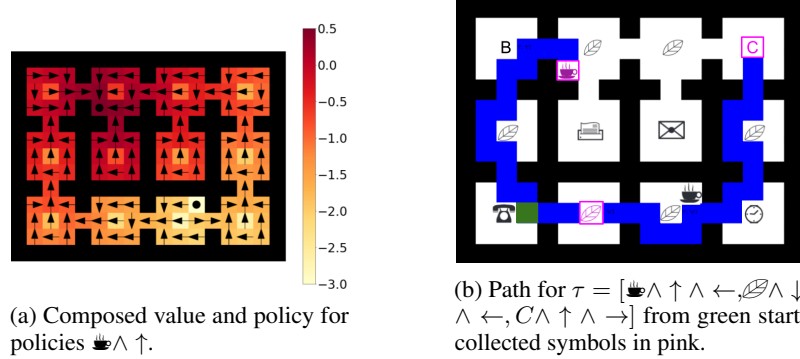

(a) Composed value and policy for policies ☕$\wedge \uparrow$.

(b) Path for $\tau = [$☕$\wedge \uparrow \wedge \leftarrow,$🍃$\wedge \downarrow \wedge \leftarrow, C \wedge \uparrow \wedge \rightarrow]$ from green start, collected symbols in pink.

Figure 6: The composed policy for ☕$\wedge \uparrow$ and the path of the accepted word $\tau$ for LTL specification $\Diamond C \wedge \Diamond($☕$\wedge \uparrow) \wedge \Diamond($🍃$\wedge \downarrow)$.

Fig. 7b shows that the agent following policy $\pi_{\blacksquare}$ does not collect another color or shape. The path obeys MV semantics as it contains no additional symbols and never violates the specification.

The second example is shown in Figure 8. For the LTL specification $\Diamond(b \wedge \neg \blacksquare) \wedge \Diamond p$, "eventually (blue and not square) and eventually purple", Comp-LTL produces the word $\tau$ $[\bullet, \blacksquare, \blacksquare]$, which corresponds to the sequence of Boolean composition policies given by $[\pi_{\bullet}, \pi_{\blacksquare}, \pi_{\blacksquare}]:=[\pi_b \wedge \pi_{\bullet}, \pi_b \wedge \pi_{\blacksquare}, \pi_p \wedge \pi_{\blacksquare}]$. The agent progresses along the list of policies in the order provided, so first the agent begins executing $\pi_{\bullet}$. Once the agent has reached a region that produces $b \wedge \bullet$, the agent transitions to executing the next policy. The agent is done when it has reached a region that produces the symbols of the final policy. In this instance, the agent is finished when it enters the region that produces $p \wedge \blacksquare$.

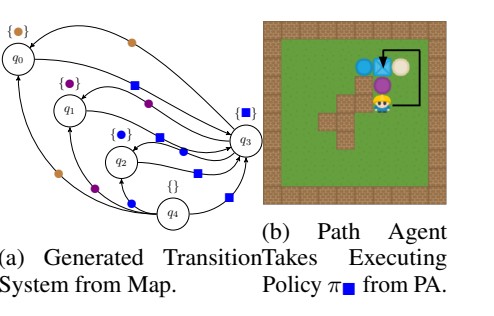

(a) Generated Transition System from Map.

(b) Path Agent Takes Executing Policy $\pi_{\blacksquare}$ from PA.

Figure 7: Pipeline for $\Diamond \blacksquare$.

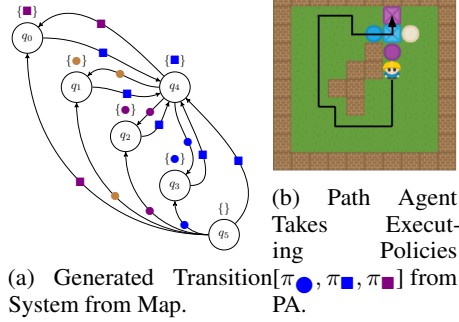

(a) Generated Transition System from Map.

(b) Path Agent Takes Executing $[\pi_{\bullet}, \pi_{\blacksquare}, \pi_{\blacksquare}]$ from PA.

Figure 8: Pipeline for $\Diamond(b \wedge \neg \blacksquare) \wedge \Diamond p$.

Fig. 8b shows that the agent following a policy does not collect another color or shape. The agent travels the long way around the center obstacle to only collect the blue circle without encountering additional symbols. Again, the path obeys MV semantics.

**Remark 1** *A trade-off of our approach is demonstrated in this case study. The agent does not take the shortest path in the environment, $\{\bullet, \blacksquare, \bullet\}$, since we only consider path length in the automaton. The paths $\{\bullet, \blacksquare, \bullet\}$ and $\{\bullet, \blacksquare, \blacksquare\}$ both have automaton path length 3. This is one of the primary trade-offs for zero-shot satisfaction, and methods such as RM can use fine-tuning to address this trade-off, but require additional training episodes.*

**Continuous Environment**   The continuous environment is a Bullet physics gym environment (Gronauer, 2022). To mirror the video game environment in a continuous scenario, each 3D object is characterized by a color and a shape from the set propositions {w (white), b (blue), p (purple), $\bullet$ (sphere), $\blacksquare$ (box)}. Optimal policies for each proposition are approximated by TD3 modified with MV safety semantics (Fujimoto et al., 2018; Leahy et al., 2024).

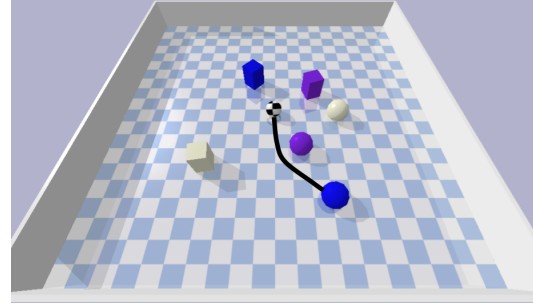

Figure 9: Comp-LTL path for $\Diamond\bullet \wedge (\neg\bullet \mathcal{U} b)$

Given an LTL specification $\Diamond\bullet \wedge (\neg\bullet\mathcal{U}b)$, our accepted word $\tau = [b \wedge \bullet]$. Figure 9 shows execution of the composed policy $\pi_{\bullet} := \pi_b \wedge \pi_{\bullet}$.

## 4.2 COMPARISON

We compare our approach, Comp-LTL, to three other state-of-the-art approaches: BC, RM, and SM. Comp-LTL trains tasks primitives using safety properties before run time and combines models temporally as needed using composition at run time (zero-shot) using environmental information to satisfy the specification. These safety-focused policies are MV policies. BC (Nangue Tasse et al., 2020) trains task primitives before run time and combines models as needed using composition. To demonstrate the necessity of safety primitive policies, we train primitive policies using BC and replace our safety primitive policies in our pipeline with their primitive policies. Table 1 highlights Comp-LTLs contributions and shows how we provide a thorough comparison suite by comparing to methods with the closest functionality.

We compare the approaches based on three metrics 1) path safety; 2) training time; and 3) specification processing time. Path safety ensures that when a primitive policy (or composition of primitive policies) is being executed, no other symbol is produced unless necessary. Training time is the time for a primitive policy to be fully trained. This is not applicable for RM as there are no primitive policies to train. Specification processing time is the time for the approach to recalculate the approach based on a new LTL specification. All comparison results for the video game environment are collected with the 13a environment configuration, to ensure a possible clear path for every specification.

| Environment | Time (s) ($\downarrow$ better) | | |
|---|---|---|---|
| | MV | BC | SM |
| Video Game | 288,160 | 218,164 | **3,398** |
| Office World | 10.70 | **8.79** | 31.30 |

Table 2: Average time to train primitive models per environment.

Table 2 shows that in the both environments, MV primitive policies take longer to train than non-MV policies for a single training pass.

| LTL Specification | Time (s) (↓ better) | | | |
|---|---|---|---|---|
| | Comp-LTL | RM (QL,CRM,RS) | RM (HRM,RS) | SM |
| $\Diamond(b \wedge \blacksquare)$ | **0.04** | 3,399.59 | 4,441.54 | 7.49 |
| $\Diamond(p \wedge \bullet)$ | **0.04** | 3,789.82 | 4,414.29 | 6.79 |
| $\Diamond b \wedge \square\neg\blacksquare$ | **0.06** | 2,192.17 | 3,004.70 | 7.46 |
| $\square(\Diamond(b\wedge\blacksquare))\wedge\square(\Diamond(p\wedge\bullet))$ | **0.03** | 18,141.99 | 47,982.74 | 7.20 |
| $\Diamond$ 🖐 | **0.02** | 5.38 | 8.73 | 0.09 |
| $\square(\Diamond ☎) \wedge \square(\Diamond ☕)$ | **0.03** | 6.76 | 10.06 | 300.13* |
| $\square(\neg B) \wedge \Diamond(\bigcirc\bigcirc🖉)$ | **0.02** | 10.06 | 12.39 | 0.12 |

Table 3: Time to reprocess given a new LTL specification (*=few-shot).

| LTL Specification | Comp-LTL | Comp-LTL + BC Policies | RM (QL, CRM,RS) | RM (HRM, RS) | SM |
|---|---|---|---|---|---|
| $\Diamond(b \wedge \blacksquare)$ | 0 | 1 | 1 | 1 | 1 |
| $\Diamond(p \wedge \bullet)$ | 0 | 0 | 0 | 0 | 0 |
| $\Diamond b \wedge \square\neg\blacksquare$ | 0 | 0 | 1 | 1 | 2 |
| $\square(\Diamond(b\wedge\blacksquare))\wedge\square(\Diamond(p\wedge\bullet))$ | 0 | 0 | 0 | 0 | 0 |
| $\Diamond$ 🖐 | 0 | 1 | 1 | 1 | 2 |
| $\square(\Diamond ☎) \wedge \square(\Diamond ☕)$ | 0 | 4 | 0 | 0 | 1* |
| $\square(\neg B) \wedge \Diamond(\bigcirc\bigcirc🖉)$ | 0 | 2 | 0 | 1 | 0 |
| Total | **0** | 8 | 3 | 4 | 6 |

Table 4: Number of additional symbols collected not required in the specification (↓ better). Green indicates the specification was satisfied by the symbols collected, red indicates unsatisfied. (*=few-shot).

Table 3 shows that upon a new LTL specification, Comp-LTL takes significantly less time to reprocess. Our training time is linear to the number of primitive task policies, but our reprocessing time does not vary greatly; however, the more complex the RM, the longer RM takes to train.

Table 4 show that Comp-LTL's additional training for safety results in no additional symbols generated other than the symbol for the primitive policy and that we are the only approach to consistently satisfy the specification. Comp-LTL with BC policies is our framework with our MV policies swapped for BC policies. Table 4 also shows that all other approaches collect multiple additional symbols. We show that Comp-LTL is also the only guaranteed zero-shot solution, as SM, the only other zero-shot capable comparator, requires additional training to satisfy one of the specifications.

## 5 CONCLUSION

We present Comp-LTL, an end-to-end zero-shot approach for executing an LTL task specification. Our pruned TS representation of the environment is deterministic, contains only feasible transitions, and is sound. Our results show that our zero-shot approach requires no additional training per specification, and the paths our approach produces are safe and feasible. While Comp-LTL has a linear cost for training the primitive task policies, our run time computation cost is minimal. Our approach agnostic to the method in which the policies are trained, as we show Comp-LTL is successful with tabular Q-learning and DRL policies in grid-based and continuous environments.

Future work includes demonstrating the effectiveness of Comp-LTL on a variety of systems, including but not limited to a environment with moving objects or changing physics (e.g., terrain friction).

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
