# OpenReview forum: "Comp-LTL: Temporal Logic Planning via Zero-Shot Policy Composition"
_ICLR.cc/2026/Conference — Submitted to ICLR 2026_

### Official Review · Reviewer_BaXj · 2025-10-31

**Soundness:** 2
**Presentation:** 2
**Contribution:** 2
**Rating:** 2
**Confidence:** 3

**Summary:**

This paper proposes an approach that composes pretrained task primitives to satisfy Linear temporal logic (LTL) specifications, with the aim of avoiding retraining or fine-tuning whenever the specifications change. The core method constructs a transition system (TS), which is then pruned and made deterministic. The approach is compared with prior work.

**Strengths:**

The approach aims to address an important problem in learning for LTL: the need to retrain or fine-tune whenever specifications change. In addition, the figures in the paper are clear and illustrative.

**Weaknesses:**

- **W1.** The related-work section is not well organized or sufficiently extensive. For example, as far as I understand, your approach is quite similar to model-based planning without learning. The works by Qiu et al. (2023) and Jackermeier & Abate (2025) also appear closely related but were only briefly mentioned and were not explicitly compared with your approach.
- **W2.** The contributions are not very clear. As I understand it, zero-shot approaches for LTL already exist; the novelty of your approach seems to stem from implicit safety integration rather than from being zero-shot per se, and this is not emphasized—for example, in the abstract.
- **W3.** In my opinion, the assumptions are very strong: the environment is modeled as a deterministic Markov decision process (MDP) whose topology can be constructed, which makes the method strongly related to model-based planning approaches such as Kurtz & Lin (2023) rather than learning approaches.
- **W4.** Only the construction of a TS is explained in the technical approach section. The pretrained task primitives, arguably the core of the approach, are neither explained nor discussed, which makes the motivation for the TS unclear as well its function.
- **W5.** The experiments are not comprehensive; only two environments and two baselines are considered.


### References
1. Kurtz, Vince, and Hai Lin. “Temporal logic motion planning with convex optimization via graphs of convex sets.” *IEEE Transactions on Robotics* 39.5 (2023): 3791–3804.

**Questions:**

- **Q1.** See W1. Could you provide a more systematic related-work discussion and comparison? Possible categories: model-based planning for LTL; model-free and model-based learning for LTL; transfer learning and fine-tuning for LTL; zero-shot transfer for LTL. Could you explicitly state the advantages and disadvantages of your work relative to model-based planning approaches? What are the advantages and disadvantages of implicit vs. explicit safety, and why is this important?
- **Q2.** See W2. Also, what does your approach contribute beyond Kloetzer & Belta (2008) and Nangue Tasse et al. (2020)?
- **Q3.** What are your thoughts on W4?
- **Q4.** Could you provide results for additional environments and compare your approach with methods beyond reward machines (RMs) and skill machines (SMs), particularly with other zero-shot and model-based planning approaches?

---

> ### Author Response · Authors · 2025-11-24
>
> We thank the reviewer for the careful review of our paper. We appreciate the detailed feedback and are looking forward to incorporating the recommended improvements into the manuscript.
>
> **Related Works (W1,Q1)**
>
> We appreciate the opportunity to explain our choice of related works and expand upon our differences to related works. Due to the three focuses of Comp-LTL--zero-shot generalization to unseen specifications, implicit safety guarantees, LTL specification satisfaction guarantees--we focus on approaches that combine at least two of those focuses. *We added a more comprehensive related works approach in Appendix B.*
>
> Qiu et al. (2023) and Jackermeier & Abate (2025) may appear closely related as they also provide zero-shot LTL specification satisfaction, but differ in two main ways:
> - Don’t compose policies, must train on every goal: Like our approach, Qiu et al. (2023) learn basic goals to execute higher level specifications but require all tasks to be trained prior to run time, as they do not compose their policies. Jackermeier & Abate (2025) leverage the structure of Buchi automata to learn policies conditioned on sequences of truth assignments, therefore, requiring all tasks be trained as they do not compose their policies. We decompose our goals an extra step and train on primitives, so at run time we execute a composed policy of two goals which reaches the intersection of the two goals and avoids all other non-goal regions.
> - Don't focus on safety guarantees: Qiu et al. (2023) and Jackermeier & Abate (2025) both consider safety at the level of LTL, therefore requiring the safety specification at training time. Our approach allows the user to explicitly define regions to avoid but importantly additionally provide non-goal region avoidance through safety-aware semantics during training. Negating a task in a safety-aware context (i.e., within the primitive policy) means that the agent will always avoid the region associated with the negated task, which aligns with LTL requirements. If safety is not included at the task level, negation/avoidance cannot be ensured at run time, only that something will be reached. Through our approach, we are able to guarantee safe behavior.
>
> *We added a brief explanation of this to the manuscript Background section.*
>
> **Contribution Novelty (W2,Q2)**
>
> We thank the reviewer for the opportunity to further define our contribution novelty. Kloetzer & Belta (2008)’s approach satisfies LTL specifications but are fully model-based planning methods; therefore, requiring a dynamics model to replicate our learned behavior. Nangue Tasse et al. (2020) generalizes to unseen combinations of primitives but does not incorporate safety nor considers satisfying LTL specifications.
>
> We are the first approach to focus on zero-shot specification adherence with implicit safety integration while integrating traditional safety requirements. We appreciate the recommendation and *revised the abstract to include more emphasis on our safety semantics.*
>
> **Assumption of Deterministic MDP (W3)**
>
> We appreciate the opportunity to clarify how our approach differs from model based approaches when considering our assumption of a deterministic MDP. Determinism is required for the theoretical guarantee to hold, but *we demonstrate empirically that it works with function approximation and continuous environments in the manuscript Results 4.1.*
>
> We provide additional functionality when compared to model-based planning approaches, such as Kurtz & Lin (2023). Kurtz & Lin (2023) can handle regions with multiple labels for LTL specifications, but require a dynamics model to provide any guarantees. We don’t require a dynamics model to provide our guarantees, as all our primitives are trained on the environment. *We added this distinction with more clarity to Appendix B (Related Work).*
>
> **Pretrained Task Primitives (W4,Q3)**
>
> The reviewer is correct that the safe primitive policy generation is not a part of our methodology. The pretrained task primitives are incorporated into our approach and linked sequentially to satisfy LTL specifications, but the primitive training is a novelty of Leahy et al. (2024). We define the safety semantics for the primitive policies in the Background, and in more detail in Appendix D.2. We describe our pretraining pipeline in Appendix H.3, but *moved a brief description of the training reward structure to the Background.*
>
> **Experiment Scope (W5,Q4)**
>
> As noted in our response to Reviewer P6SD, our experiments were rooted to the environments of our baseline comparisons. We compare to baselines that cover the span of related approach functionality categories, see Table 1, which we believe is a sufficient comparison scope. The four approaches we compare to (Comp-LTL+BC policies, RM+QL, RM+HRM, and SM) are the closest comparisons in terms of capability and implementation.
>
> *We show our approach in continuous environments but believe our baseline suite is sufficient for comparisons.*

---

### Official Review · Reviewer_P6SD · 2025-10-31

**Soundness:** 4
**Presentation:** 4
**Contribution:** 3
**Rating:** 4
**Confidence:** 3

**Summary:**

The paper proposes a logic-based control framework that is _agnostic to the specific logical specification_, in the sense that changing the task objective, expressed as an LTL formula, does not require retraining any policy. This is achieved by combining a pre-trained set of primitive policies with a transition system (TS) that abstracts the environment into labeled regions. The TS is then composed with the automaton corresponding to the target LTL formula, inducing a product graph over which planning is performed. Consequently, adaptation to a new specification is achieved zero-shot, purely through symbolic planning on this graph, without additional policy learning.

**Strengths:**

- The paper is well written, and the contribution is clear.
- The proposed idea is novel. In particular, I find interesting the connection between temporal logic specification and zero-shot composition of policies for the similarity with the multi-task problem.
- The authors formally proove the soundness of the propose pruned transition system (TS).
- While in part I find the empirical evaluation limited, it shows promiseing results, and highlight that the the proposed framework can satisfy new logic specifications without retraining.

**Weaknesses:**

- The main concern is scalability. Constructing the transition system in realistic or continuous domains is likely intractable, especially since it requires identifying the regions where each atomic proposition holds.
- The need to train one policy for each atomic proposition $\sigma \in \Sigma$ does not scale well as the number of labels grows.
- As already anticapted, emprical evaluation is limited to toy domains. It is difficult to understand whether this method can be applied in more realistic environments.
- The framework assumes that the available primitives are sufficient to cover all relevant behaviors, which is a strong assumption in realistic settings (more on questions).

**Questions:**

- How computationally demanding is it to construct the transition system in real environments, particularly regarding the recognition of regions where specific propositions hold?
- Since the number of primitive policies required scales with the number of labels $\Sigma$, how does this behave in large or continuous domains? Could approximations or hierarchical abstractions make this approach feasible in practice?
- Also, suppose a simple navigation problem in a 2D environment. A goal can be any point (x, y), this means that I need infinite primitives to account for each goal?

Overall I find the idea interesting, my concerns are in the scalability and applicability of the method. I am open to discussion with the authors on the points raised.

---

> ### Author Response · Authors · 2025-11-24
>
> We thank the reviewer for the careful review of our paper. We appreciate the detailed feedback and are looking forward to incorporating the recommended improvements into the manuscript.
>
> **Scalability to Realistic or Continuous Domains**
>
> We appreciate the opportunity to clarify how our approach will scale to realistic or continuous environments.
>
> As noted in our response to Reviewer 6BcS, our problem formulation is not exclusive to grids, as all we require for creating the transition system (TS) is the topology of a labeled environment. Fig. 3 is meant to suggest how a continuous environment can be abstracted to a TS.
>
> The existence of a labelling function for a continuous environment suffices to train primitive policies and create a TS. Automatic label generation is its own active area of research (see [1]), so we focus on using prespecified labeling functions on a known environment. Our problem, like others in the literature, such as Jothimurugan et al. (2021) and Qiu et al. (2023), begins with a labeled map of the environment. *We extended our experiments to continuous spaces in the updated manuscript Results.*
>
> [1] Maggio et al. “Clio: Real-time Task-Driven Open-Set 3D Scene Graphs,” 2024.
>
> Regarding planning for simple navigation, we do not need infinite coordinates. We focus on region level task objectives, and each region is a closed subset of the environment: $R \subseteq E$. All of the points within that region satisfy the propositions that label it.
>
> **Limitation of Primitive Policies per Atomic Proposition**
>
> We thank the reviewer for the opportunity to expand upon the cost of policies per atomic proposition. We assert that composition, and the pretraining required, are better than training for every specification.
>
> The complexity of our training time is driven primarily by the number of primitives ($\sigma \in \Sigma$), and is not affected by the LTL specification. A more complex environment would create a larger TS and a more complex specification would create a larger Buchi automaton, which would in turn create a significantly larger product automaton (PA). However, producing a solution reduces to a search problem on a (potentially quite large) graph, which we would expect to remain significantly faster than re-training. We have found through testing that the time for functions over the PA to compute are negligible when compared to the primitive training time. We assert it is worth the upfront cost of pre-training primitives $\sigma \in \Sigma$ to have a fully zero-shot (therefore specification agnostic) safe solution.

---

### Official Review · Reviewer_6BcS · 2025-11-01

**Soundness:** 3
**Presentation:** 2
**Contribution:** 2
**Rating:** 4
**Confidence:** 4

**Summary:**

The paper introduces Comp-LTL, a framework for zero-shot satisfaction of Linear Temporal Logic (LTL) specifications using pretrained RL task primitives. Instead of retraining policies for new specifications, Comp-LTL composes existing primitives via Boolean task algebra and a pruned, deterministic transition system (TS) that ensures feasible and sound planning. The method integrates safety through minimum-violation (MV) semantics and constructs a product automaton with Büchi automata for execution. Experiments in grid-based environments (Office World, Video Game) show that Comp-LTL achieves safer, faster, and more generalizable performance than baselines such as Reward Machines, Skill Machines, and Boolean Composition.

**Strengths:**

1. The integration of deterministic TS pruning and Boolean policy composition for zero-shot LTL satisfaction is original and well-motivated.
2. The paper provides clear theorems (determinism, soundness, feasibility) with proof sketches, demonstrating a solid theoretical foundation.
3. Comp-LTL requires no fine-tuning or retraining, showing strong adaptability to unseen specifications in grid-world environments.

**Weaknesses:**

1. The Q-learning algorithms used in the paper are value-based, discrete-action algorithms — meaning they assume a finite, enumerable action space. Authors claim that "Our approach agnostic to the method in which the policies are trained, as we show Comp-LTL is
successful with both tabular Q-learning and DQN policies." but the the Q-learning-based primitives in Comp-LTL cannot be directly applied to continuous-action robotic tasks. The experiments are confined to grid-based environments. Claims of generality would benefit from evaluation in continuous control or robotic settings.
2. While runtime and training times are reported, theoretical or empirical analysis of computational complexity (e.g., TS construction scaling with number of regions or propositions) is missing.
3. The claimed contribution of abstracting a geometric environment into a transition system with Boolean-composed task labels is not novel — similar abstractions were used in "Compositional RL from Logical Specifications" (NeurIPS 2020) and "Instructing Goal-Conditioned RL Agents with Temporal Logic Objectives" (NeurIPS 2023). The more distinct contribution lies in the pruning strategy ensuring deterministic, feasible TS representations and its integration with zero-shot safe composition.

**Questions:**

1. How would Comp-LTL perform on LTL formula having "Until" operator? And how about the performance on LTL with $\omega$-regular expression, which is a very important extension on LTL formula?
2. Would Comp-LTL still maintain zero-shot in continuous-action environments?
3. See above.

---

> ### Author Response · Authors · 2025-11-24
>
> We thank the reviewer for the careful review of our paper. We appreciate the detailed feedback and are looking forward to incorporating the recommended improvements into the manuscript.
>
> **Generalization to Continuous Environments while Maintaining Zero-Shot Capability (W1, Q2)**
>
> We appreciate the opportunity to clarify how our approach will scale to realistic or continuous environments while remaining zero-shot. Our problem formulation is not exclusive to grids. All we require for creating the transition system (TS) is the topology of a labeled environment. We focused on grid environments for simplicity of presentation. Fig. 2 is meant to suggest how a continuous environment can be abstracted to a TS. As long as we have a labelling function for casting a continuous environment to labels, we would be able to train primitive policies using that function and create our TS using the labeling function.
>
> The composition method of Leahy et al. (2024) holds in continuous spaces, and nothing in our methodology precludes its use in continuous spaces. The LTL specifications were somewhat limited in complexity as we emphasized missions that were well-suited to implementation with RM (Icarte et al., 2018) to ensure a fair assessment. We are confident Comp-LTL will not degrade with more complex specifications and would continue to not require any fine tuning (remaining zero-shot).
>
> *We added additional results for a continuous space to the manuscript.*
>
> **Larger Environment Impact and Evaluation (W2)**
>
> A larger grid world or continuous space would not impact Comp-LTL more than other approaches. The complexity of our training time is driven primarily by the number of primitives, for both grid-based and continuous domains, and is not affected by the LTL specification. Other methods, such as BC (Tasse et al., 2020), (Leahy et al., 2024), and SM (Tasse et al., 2024) would scale similarly with the environment, while methods such as RM (Icarte et al., 2018) are also impacted by the specification itself. Roughly, RM scales according to the size of the specification (which is encoded as the machine), whereas our approach, Comp-LTL, and SM scales with the number of primitives.
>
> Larger environments and complex specifications would increase the size of the transition system, Buchi automaton, and product automaton, of course impacting the computation time of a new solution. The product automation (PA) generation, which is affected by the LTL specification, is independent of the training process (and is therefore how we are zero-shot). Producing a solution reduces to a search problem on a (potentially quite large) PA graph, which we would expect to remain significantly faster than re-training.
>
> **Novelty of Abstracting Transition System into Regions with Multiple Labels (W3)**
>
> We appreciate the opportunity to clarify the novelty of our contribution. We claim to contribute “a method for abstracting a geometric representation of an environment into a transition system (TS) with transition labels representing feasible Boolean combinations of tasks to transition between multiply labeled regions.” Our novelty lies in manipulating a TS to have the deterministic qualities we require in order to make our guarantee claims.
>
> We differ from "Compositional RL from Logical Specifications" (NeurIPS 2020) and "Instructing Goal-Conditioned RL Agents with Temporal Logic Objectives" (NeurIPS 2023) in notable ways. Neither work abstracts their environment as a TS, instead operating at the level of the specification (i.e., on the automaton encoding the specification). They do not account for the environment topology. Thus, their ability to generalize is limited to specific automaton constraints they have encountered in training, whereas our approach uses the environment topology to inform the choice of feasible composed policies.
>
> **Additional Operators and $\omega$-Regular Languages (Q1)**
>
> Our approach supports the Until operator, as it can be represented in an automaton. The semantics of symbol production, i.e., if empty regions can violate Until, is a consideration at training time.
>
> Our approach supports $\omega$-regular languages. The minimum violation (MV) semantics from Leahy et al. (2024) consider finite traces. However these semantics are for concurrent composition. In our work, we concatenate these finite traces into (potentially infinite) sequences, in a temporal composition. Thus, the Boolean composition, and its associated finite traces, corresponds to traversing an edge of the Buchi automaton. An infinite sequence in the Buchi automaton would correspond with an infinite sequence constructed by concatenating an infinite sequence of finite traces

---

### Official Review · Reviewer_Temd · 2025-11-01

**Soundness:** 1
**Presentation:** 2
**Contribution:** 2
**Rating:** 2
**Confidence:** 3

**Summary:**

This work considers the problem of composing task primitives using LTL specifications. This is an established approach within the literature and this work aims to extend on this in two ways: 1) by incorporating an LTL pruning mechanism which simplifies the transition system defining the temporal sequence of tasks 2) by incorporating safety into the primitives themselves rather than relying on the LTL specification to guide the safety concerns which is done in prior work. The paper compares their approach to the state of the art approaches to using LTL specifications for task composition and shows that their approach is superior on a safety metric and in terms of learning speed.

**Strengths:**

## Originality
The work is grounded quite closely to the literature on using LTL specifications for temporal and spatial task composition. This is not inherently bad, and in fact by positioning the work clearly against these prior works it does make the differences stand out.

## Quality
The motivation of the work is clear and the hypothesis is grounded in prior work. The results that are presented are interpreted fairly.

## Clarity
The paper is well written and figures are clear and useful overall. The paper uses notation and symbols in a way that is typical of this line of work which makes it easier to follow.

## Significance
The work considers an important problem - safety within RL and also does support faster learning which is important as we expand our models into more difficult domains. Thus, there could be future work which builds on this paper and its stated claims.

**Weaknesses:**

## Clarity
Firstly the minor concern, the figure captions are very uninformative and this limits the benefit of the figures substantially. Figure 3a in particular really needs to be more descriptive both in terms of the caption and figure itself. The work also uses jargon which is not sufficiently defined such as "sound". When a word is used in this manner to mean something technical it is necessary to define it formally.

More importantly, it is very difficult for me to see the connection between the two main concerns of this work: the TS pruning and the approach of embedding safety directly into the primitives. This seems like two entirely distinct directions and makes the overall structure of the paper confusing.

## Quality and Significance
My first critique here is that the paper takes for granted that safety should be embedded directly into the task primitives rather than specified in the LTL. This is not obvious to me and undermines the entire direction as a result. I would greatly appreciate clarity on why we even want this in the first place. Secondly, the consequences of putting the safety behaviour into the training of the primitives is not given due consideration. My understanding is that this will make all of the task primitives sub-optimal and inflexible to cases where the constrains may be temporary. So while the domains with fixed constrains may be fine for this, the flexibility of the approach is limited greatly and by extension so is the applicability of the model. Remark 1 similarly notes a trade-off which emerges from the paper's approach to zero-shot satisfaction and the possibility of introducing sub-optimality and notes that RMs take a different approach by fine-tuning. But then why is the paper phrased as if it improves on RMs and SMs in this regard (for example on lines 471 to 473)? What is the point of being "full-zero shot" if the proposed method is also suboptimal at this just like the prior work which at least considers fine-tuning.

Finally, Table 4 seems unreasonable to me and is poor experimental design. Perhaps I am missing something, but to compare the prior methods on a fairly arbitrary metric (number of additional symbols collected) which they were not trained to consider at all, while the proposed method explicitly optimises the metric and then claims to be superior, is fairly meaningless. I would appreciate more explanation on why this is even a fair comparison.

**Questions:**

I have asked some question in the review above which I would appreciate answers to. Additionally, what is the connection between the TS pruning and safety? Why report the computation time and the safety metric if this work is primarily concerned with safety? How should I interpret the speed-up relative to prior work when there is a trade-off as a result of this speed up (Remark 1)?

---

> ### Author Response · Authors · 2025-11-24
>
> We thank the reviewer for the careful review of our paper. We appreciate the detailed feedback and are looking forward to incorporating the recommended improvements into the manuscript.
>
> **Additional Definitions and Figure Descriptions**
>
> We thank the reviewer for noting figures and concepts we should define more thoroughly. We added a definition for safety. We clarified Figure 3 by expanding 3a's caption and adding colors to show which regions in 3a correspond to states in 3b.
>
> **Connection Between Transition System (TS) Pruning and Embedding Safety into Primitives**
>
> We appreciate the opportunity to clarify a key aspect of our approach. Embedding safety in the tasks is prior work, but the TS generation and subsequent pruning in conjunction with using safe primitives is unique. *We have clarified this in at the end of the Introduction.* Our goal is to produce a sequence that is guaranteed to satisfy an LTL specification. Neither embedding safety in the policies nor pruning the TS on its own will accomplish this.
>
> The TS is a representation of the environment, and we prune the TS to be deterministic and feasible (i.e. no impossible transitions) with respect to the learned policies. The Buchi automaton is a representation of the LTL specification. Taking the product between the two, in the form of a Product automaton (PA), ensures any output from the PA is both feasible and deterministic given our environment. The PA provides the sequence of safe policies to execute ($\tau$) in order to satisfy the LTL specification in the given environment. Therefore, using our pruned TS in the PA ensures we can feasibly and deterministically satisfy the LTL specification in our given environment topology while the primitive policies ensure the execution of the policies to adhere to the specification are safe. Figure 2 represents how the two are combined. For more information on the product construction please refer to Appendix F.
>
> Imagine an environment with five rooms connected in a line: kitchen, living room, hallway, bedroom, kitchen, and the specification "! bedroom U kitchen". If an agent is in the hallway, and we use a policy trained on "kitchen", the policy will either go through the living room or the bedroom according to MV semantics. Since the dynamics are unknown, it is not clear which path will be followed, despite safety being embedded into the policy. By pruning the TS (in this instance, case 2), the ambiguity is resolved and the specification can be satisfied.
>
> **Safety in Task Primitive Policies**
>
> We thank the reviewer for the opportunity to expand on why we desire an approach with safety embedded into the primitive policies.
>
> Other compositional works consider reachability-only (RO) semantics, whereas we consider safety-aware semantics, defined by Leahy et al., 2024. Negating a task in RO context means an agent will not terminate in the region associated with the negated task, but it could pass through that region. Negating a task in a safety-aware context (i.e., within the primitive policy) means that the agent will always avoid the region associated with the negated task, which aligns with LTL requirements. If safety is not included at the task level, negation/avoidance cannot be ensured at run time, only that something will be reached. Other approaches that use safety at the level of LTL require the safety spec at training time, but since we desire a zero-shot approach we don't train for only a single spec. *We added this explanation to the manuscript Background.*
>
> **Zero-shot Tradeoff**
>
> The increased training time up front and the suboptimal execution paths (Remark 1) are the cost of zero-shot guaranteed specification satisfaction and no training or fine tuning for new specifications.
>
> **Experimental Design**
>
> We acknowledge that our experiment environments may appear limited, but our goal with these experiments was to emphasize the need for safety-aware policies by comparing to Tasse et al. (2020)’s reachability-only policies, the value of specification-agnostic training by comparing to Icarte et al. (2018), along with the associated trade-offs between training and runtime. Therefore, we selected environments and specifications that were used in Icarte et al. (2018) and Tasse et al. (2020) to provide the fairest comparison possible. There are no other methods to directly compare our approach to as we are the first group to consider safety-aware primitive policies when executing the output of a PA which satisfies an LTL specification.
>
> Table 4 emphasizes the point about the need for safety-aware policies and semantics when executing paths to satisfy an LTL specification. Since all approaches are learning-based, the user defining the LTL specification does not have insight into exactly what paths will be taken given a new environment configuration. Therefore, incorporating safety-aware policies which reduce extraneous symbol generation allows for the path to be as safe as possible.

---

### Author Response · Authors · 2025-11-24
**Main Points**

While we provide detailed comments to the reviewers below, we would like to address the primary criticisms briefly here, as well as highlight changes to our updated manuscript.

**Contributions**

We develop a method for abstracting a geometric representation of an environment into a transition system (TS) with transition labels representing feasible Boolean combinations of tasks to transition between regions with multiple labels, and we resolve nondeterminism in the transitions enabled by the Boolean composition of safe primitive task policies.

Embedding safety in the tasks is prior work, but the TS generation and subsequent pruning in conjunction with using safe primitives is unique. Our goal is to produce a sequence that is guaranteed to satisfy an LTL specification. Neither embedding safety in the policies nor pruning the transition system on its own will accomplish this.

We demonstrate that our method, Comp-LTL, allows zero-shot satisfaction of LTL specifications at run time, and the resulting behavior is inherently safe without adding specific safety criteria into the specification.

**Related Works**

We recognize the need for an expanded related works section, as our approach lies in the intersection of  of temporal logic planning, reinforcement learning, and compositional
methods for skill reuse. We have since added Appendix B to review relevant literature in these areas and highlight how our approach differs from existing methods.

**Assumptions and Scalability**

We use deterministic MDPs for theoretical guarantees but demonstrate on non-deterministic MDPs and with function approximation. Further, we are not limited to grid worlds and have demonstrated our approach in a continuous environment. This approach scales similarly to others in terms of environment size, but offers advantages in training time compared to longer specifications in automata-based training.

**Manuscript and Appendix Updates**

All updates below are denoted in green.

We have updated the manuscript to include:
- Abstract with more of a focus on the safety-aware semantics and how those safe policies integrate into the TS
- Background with addition of reachability-only (RO) semantics versus safety-aware semantics and brief description of safety-aware policy training
- Redesign of Figure 3 to ease understanding of parsing an environment into a TS
- Results with the Continuous environment
- We are currently training more primitives to show results for a more complex specification
- Changing “fully zero-shot” to the more accurate “guaranteed zero-shot”

We have updated the Appendix to include:
- A thorough literature review in Appendix B
- Environment information for the continuous domain including reward values (in Appendix H.3)
- Training information for TD3 in the continuous environment including hyperparameters (in Appendix H.3.1)

---

### Author Response · Authors · 2025-12-03
**AC Summary**

We thank the AC in advance for the review of our submission and the additional work it requires as a result of the API bug. To aid in the review, we provide a brief summary below of the reviewer identified strengths and how we address each reviewer identified weakness in our updated manuscript. We summarize our response to the reviewers along with all our manuscript and appendix updates in our Official Comment “Main Points.”

We note that our theoretical claims, methodology, and results received positive remarks. The majority of reviewers desired more experimental results, particularly in a continuous setting. We have since updated the manuscript with results for a continuous environment, to show our claims hold. The remaining remarks we were able to resolve through clarification and minor edits to the main manuscript with additional details pulled from the Appendix.

**Contribution**

We develop a method, Comp-LTL, for satisfying a temporal logic specification in a zero-shot manner using Boolean composition of primitive policies. Our method abstracts a geometric representation of an environment into a transition system (TS) with transition labels representing feasible Boolean combinations of tasks to transition between regions with multiple labels, and we resolve nondeterminism in the transitions enabled by the Boolean composition of safe primitive task policies. We demonstrate that Comp-LTL achieves zero-shot satisfaction of LTL specifications at run time, and the resulting behavior is inherently safe without adding specific safety criteria into the specification.

**Reviewer Identified Strengths**

Clear and fair writing
* The figures are clear and useful. (Temd, BaXj)
* The paper is well written and our contribution is clear. Comparing our approach clearly against prior work makes our differences stand out. (P6SD, Temd)
* We use notation and symbols in a way typical of LTL focused approaches which makes our methodology easier to follow. (Temd)
* Generally, the results that are presented are interpreted fairly. (Temd)

Clear motivation
* The motivation of our work is clear and the hypotheses is grounded in prior work. (Temd)
* The integration of deterministic TS pruning and Boolean policy composition for zero-shot LTL satisfaction is original and well-motivated. (6BcS)

Strong theoretical foundation
* We provide clear theorems (determinism, soundness, and feasibility) along with proof sketches, which demonstrates a solid theoretical foundation. (6BcS, P6SD)

Significance
* Our work is significant as it considers safety within RL and faster zero-shot processing, which is important as models expand into more complex domains. (Temd, 6BcS)
* Our approach addresses a significant problem in learning for LTL: the reliance on retraining or finetuning upon a new specification. (BaXj)

**Improvements to Address Reviewer Concerns**
* To address a need for deeper discussion of related works with a focus on model based approaches, we added a thorough literature review in Appendix B. (BaXj)
* To clarify our assumptions, primarily on deterministic MDPs and how that impacts our approach, we note to Reviewer BaXj that determinism is required for the theoretical guarantees to hold, but our initial submission demonstrates empirically that it works with function approximation in discrete environments in the manuscript Results 4.1 Video Game Environment. We added Continuous Environment results to Results 4.1 to show it works with function approximation in continuous environments. (BaXj)
* To clarify the connection between TS pruning and embedding safety into primitives, we updated the Abstract to focus on the safety-aware semantics and how those safe policies integrate into the TS. We also updated the Introduction to note how both pruning the TS and embedding safety into the primitives are required to guarantee LTL specification satisfaction. (Temd)
* To elucidate the reason to integrate safety into task primitive policies, we added the notion of reachability-only (RO) semantics and safety-aware semantics, defined by Leahy et al., 2024, to the Background, to highlight how embedding safety into primitives ensures LTL specification satisfaction, along with a brief description of safety-aware policy training. (Temd)
* To show how our approach scales to more complex domains, we added results for a continuous environment and are currently training more primitives to show results for a more complex specification. We added environment information for the continuous domain including reward values to Appendix H.3 and added training information for TD3 in the continuous environment including hyperparameters to Appendix H.3.1. (6BcS, P6SD, BaXj)

---

### Meta-Review · Area_Chair_8phg · 2026-01-06

**Summary:**

The paper proposes Comp-LTL, a framework for achieving zero-shot satisfaction of Linear Temporal Logic (LTL) specifications in reinforcement learning (RL) agents.
Specifically, the paper proposes to decouple the learning of “task primitives” from the LTL specification. They do so by first pre-training a set of safety-aware primitive policies and then composing them. In more detail, they convert the geometric environment into a Transition System (TS) where transitions represent feasible Boolean compositions of primitives, they then prune this TS to remove unrealizable transitions and combine the pruned TS with the Buchi automaton corresponding to the given LTL specification.
This design allows the agent to satisfy new specifications in a zero-shot fashion, without retraining, unlike prior work that requires retraining for each new LTL formula.

The main concerns that the reviewers raised are the following:

- *C1*. Clarity issues pertaining to figure captions, jargon used without explanation, unclear connection between two key considerations: TS pruning, and the design choice to embed safety directly into the primitives (Reviewer Temd). Clarity issues with respect to related work (Reviewer BaXj) making it hard to assess novelty over prior work (Reviewer BaXj). Pretraining primitives are not discussed (Reviewer BaXj).
- *C2*. The design choice that safety should be embedded into the primitives is questionable and needs justification. The authors should justify why this is desired and what are the drawbacks or consequences of this approach (it might make all task primitives unnecessarily suboptimal and is at best inflexible with respect to evolving safety constraints) (Reviewer Temd).
- *C3*. Questionable metric (number of additional symbols collected) that may favour the given method since it directly optimizes for them whereas other methods don’t (Reviewer Temd).
- *C4*. Limitation to discrete action spaces. While the authors claim their approach is method agnostic, it relies on value-based Q-learning, where actions are finite and discrete (Reviewer 6BcS, Reviewer P6SD).
- *C5*. Missing analysis of computational complexity and scalability of the construction of TS with the number of regions or atomic propositions (Reviewer 6BcS).
- *C6*. The novelty of abstracting a geometric environment into a TS with boolean-composed task labels is overclaimed (relevant papers were missed) (Reviewer 6BcS).
- *C7*. Evaluation limited to toy domains / hard to assess applicability to realistic environments (Reviewer P6SD); only two environments and two baselines considered (Reviewer BaXj).
- *C8*. Concern of scalability of primitive construction / number of policies we need to train  (Reviewer P6SD).
- *C9*. Unrealistic assumption that the available primitives are sufficient to cover all relevant behaviors  (Reviewer P6SD).
- *C10*. Very strong assumption that the environment is a deterministic MDB whose topology can be constructed.

**Reviewer Concerns:**

The paper has several strong points, like a strong theoretical foundation, which the reviewers appreciated. The integration of deterministic TS pruning with boolean composition seemed original and well-motivated to some reviewers, and the paper makes an interesting connection between temporal logic and zero-shot composition. Some reviewers noted  that some explanations in the paper and figures are clear and easy to understand.

However, certain concerns remain unaddressed, or partially addressed, including: missing formal complexity analysis for construction of the TS, missing an explicit cost-benefit analysis of upfront training of the primitives to be "worth it" for zero-shot capabilities, to understand when the trade-off becomes positive. Other limitations refer to the use of toy domains (though the authors added an additional continuous environment which is great) and questionable metric for comparison with prior work. An important concern also relates to clarity when it comes to the differentiation of this method to prior work and where its novelty lies, as well as why different design choices were made. While the authors have added discussions during the rebuttal, the paper could benefit from another round of review to integrate this information.

**Reviewer Scores:**

**Reviewer Temd**


The authors responded to C1 by making updates to the revised paper to improve clarity. When it comes to the connection between the two key considerations, the authors argue that the design choices of embedding safety into primitives and TS pruning work together to achieve the goal of producing a sequence that is guaranteed to satisfy an LTL specification. Neither of the two alone will accomplish this. This makes sense, within the context of their solution, but I’m wondering if there is a clearer way to give the reader an intuition of what other combinations exist and why those would fail. It may help to mention all of the desiderata upfront, what are the different design choices at play and what are the trade-offs between them.


To address C2, the authors argued that without embedding safety into the primitive policy, they cannot guarantee safety at runtime for negated constraints, explaining a distinction between RO semantics and safety-aware semantics.The authors conceded that suboptimality is the cost to be paid for guaranteeing their goal can be met.


To address C3, the authors argue that reducing extraneous symbol generation allows for paths to be as safe as possible, which if I understand correctly seeks to imply that this is the correct quantity to optimize for, and to evaluate on. But I’m not convinced that minimizing the number of additional collected symbols is necessary for safety in all cases (at least it would help to elaborate on this), and the fact remains that their method may have an unfair advantage on this metric as it has been explicitly optimized for it.


Based on the fact that C1 and C3 are only partially addressed, I doubt that the reviewer (who initially recommended rejection) would increase their score.


**Reviewer 6BcS**
To address C4, the authors clarify that their formulation relies only on the topology of a labeled environment, not on discrete grids. To show empirically that their method is not limited to discrete action spaces, they added new results for a continuous environment to the manuscript, using TD3 (a continuous control algorithm) instead of Q-learning, where they show successful execution. This response addresses C4 well.


To address C5, while the authors don’t offer analysis of computational complexity, they discuss the scalability of their methods and others at an intuitive level, with respect to the size of the grid (or continuous space) versus the number of primitives, clarifying that the main driver of complexity is the latter for their approach. This explanation is reasonable at a qualitative level but a formal analysis is missing.


To address C6, the authors clarify their differentiation from the other papers that the reviewer brought up, which do not leverage the environment topology. This does offer a clear differentiation between this work and prior works, though I’m not sure the reviewer would find this significant enough.


Overall, some of the reviewers comments were addressed sufficiently (C4, maybe C6) while others partially (C5). The reviewer previously recommended a weak rejection and may maintain their opinion given some unresolved issues.


**Reviewer P6SD**

They respond to the reviewer’s concern about scalability to continuous domains by reiterating the response to Reviewer 6BcS, and the new experiments. They also highlight that automatic label generation is a research area in its own right, so this work focuses on using prespecified labeling functions.

To address C8, the authors claim that the up-front cost paid to train the policies is worth it for then having a zero-shot solution. But there is no explicit experiment (or compilation of existing results) to support this with concrete empirical evidence like “Our method pays off after X tasks”.

As far as I can tell, the authors don’t attempt to answer C9 directly (though maybe it’s indirectly addressed because of the assumption of starting from a prespecified environment, which would offer some clarification but does not alleviate the concern).

Overall, the response partially addresses the comments but I’m not certain that the reviewer (who recommends weak rejections) would have bumped their score to an acceptance in light of these responses.


**Reviewer BaXj**

To address clarity issues around the relationship between this work and related work and novelty, the authors have added extensive discussion. In my view, this does differentiate between different categories clearly enough, like model-based planning that the reviewer referred to. The authors then specify that their novelty comes from proposing the first approach that achieves zero-shot specification adherence with implicit safety integration while integrating traditional safety requirements. While this seems true, I’m not sure that the reviewer would find this substantially different nor that the motivation for tackling this particular variant is clear enough.

To address C10, the authors clarify that determinism is needed for their theoretical results but empirically they demonstrate that their method works with function approximation and continuous environments, which is convincing.

They also clarify the scope of their method (in particular that the safe primitive policy generation approach is reused from prior work), and that their experimental settings were fixed to those in prior papers for fair comparisons with prior work that they claim is comprehensive.

Overall, I doubt that this reviewer (who initially recommended rejection) would substantially change their score, given weaknesses in writing quality that make it hard to assess where the contribution and novelty arise from (which have been to some degree addressed in the revisions, but a cleaner reorganization for another round of review may help), and limitations in experimental scope.

---

### Decision · Program_Chairs · 2026-01-26

Reject